# Polyamine Catabolism in Acute Kidney Injury

**DOI:** 10.3390/ijms20194790

**Published:** 2019-09-26

**Authors:** Kamyar Zahedi, Sharon Barone, Manoocher Soleimani

**Affiliations:** 1Departments of Medicine, University of Cincinnati College of Medicine, Cincinnati, OH 45267, USA; Sharon.barone@uc.edu; 2Research Services, Veterans Affairs Medical Center, Cincinnati, OH 45220, USA; 3Center on Genetics of Transport, University of Cincinnati College of Medicine, Cincinnati, OH 45267, USA

**Keywords:** acute kidney injury, polyamine, spermidine, spermine, polyamine catabolism, spermine/spermidine N^1^-acetyltransferase, spermine oxidase, H_2_O_2_, 3-aminopropanal, cell cycle, innate immune response, endoplasmic reticulum stress/unfolded protein response, mitochondria, lysosome, diminazene aceturate, MDL72527, phenelzine, catalase

## Abstract

Acute kidney injury (AKI) refers to an abrupt decrease in kidney function. It affects approximately 7% of all hospitalized patients and almost 35% of intensive care patients. Mortality from acute kidney injury remains high, particularly in critically ill patients, where it can be more than 50%. The primary causes of AKI include ischemia/reperfusion (I/R), sepsis, or nephrotoxicity; however, AKI patients may present with a complicated etiology where many of the aforementioned conditions co-exist. Multiple bio-markers associated with renal damage, as well as metabolic and signal transduction pathways that are involved in the mediation of renal dysfunction have been identified as a result of the examination of models, patient samples, and clinical data of AKI of disparate etiologies. These discoveries have enhanced our ability to diagnose AKIs and to begin to elucidate the mechanisms involved in their pathogenesis. Studies in our laboratory revealed that the expression and activity of spermine/spermidine N^1^-acetyltransferase (SAT1), the rate-limiting enzyme in polyamine back conversion, were enhanced in kidneys of rats after I/R injury. Additional studies revealed that the expression of spermine oxidase (SMOX), another critical enzyme in polyamine catabolism, is also elevated in the kidney and other organs subjected to I/R, septic, toxic, and traumatic injuries. The maladaptive role of polyamine catabolism in the mediation of AKI and other injuries has been clearly demonstrated. This review will examine the biochemical and mechanistic basis of tissue damage brought about by enhanced polyamine degradation and discuss the potential of therapeutic interventions that target polyamine catabolic enzymes or their byproducts for the treatment of AKI.

## 1. Introduction

Acute kidney injuries (AKI) are marked by an abrupt decrease in kidney function that may be caused by damage to the renal parenchyma or abrupt reduction in renal perfusion [1,2,3]. These injuries are complex disorders that can be caused by any of multiple causes, occur in a variety of settings, and have diverse clinical manifestations that range from a small but sustained elevation in serum creatinine to anuric renal failure [1,2,3]. Approximately 7% of all hospitalized patients and nearly 35% of intensive care patients suffer from AKI [3]. Mortality from AKI remains high, particularly in critically ill patients, where it can be more than 50% [3]. The primary causes of AKI include ischemia/reperfusion (I/R), sepsis, or nephrotoxicity (e.g., contract dyes, aminoglycosides, or cisplatin); however, AKI patients may present with a complicated etiology where many of the aforementioned conditions co-exist [1,2,3]. To date, examination of AKI of differing etiologies have identified multiple bio-markers that enhance our diagnostic abilities, and many molecules and pathways that further our understanding of its pathogenesis. Despite these advances, the main option for the treatment of AKI remains the provision of supportive care and renal replacement therapy. The diversity of instigating factors and complex etiology of AKI may be the primary reason for the lack of treatment modalities and suggest that multiprong approaches are needed for therapeutic interventions to treat these injuries.

During the process of identifying AKI biomarkers, it was determined that the expression and activity of spermine/spermidine N^1^-acetyltransferase (SAT1), the rate limiting enzyme in polyamine back conversion, was elevated in kidneys subjected to I/R injury [4]. In addition to SAT1, the expression of spermine oxidase (SMOX), another important enzyme in polyamine catabolism, is also increased in kidneys of animals subjected to I/R-induced AKI [4,5]. The maladaptive role of polyamine catabolism has been established in a broad range of acute injuries [4,6,7,8,9,10]. This review will examine the role of polyamine catabolism in the mediation of various acute tissue injuries with a specific focus on AKI. In addition, it will focus on the biochemical and mechanistic basis of tissue damage resulting from enhanced polyamine degradation and the potential for therapeutic interventions targeting polyamine catabolic enzymes or their byproducts for the treatment of AKI.

## 2. Regulation of Polyamines and the Polyamine Metabolic Pathway

Polyamines, putrescine, spermidine, and spermine are cationic aliphatic amines that are important in the regulation of interactions between biological macromolecules (e.g., nucleic acid–nucleic acid, nucleic acid–protein, and protein–protein interactions). In cells, a significant fraction of polyamine molecules is bound to RNA and play important roles in the regulation of protein translation [11,12]. Changes in polyamine levels affect the structure of the DNA double helix [13,14], modify nucleic acid (DNA or RNA) interactions with proteins [15,16], affect enzymatic activity, and protein–protein interactions [17,18,19]. Because of their important biological roles, the cellular levels of polyamines are tightly regulated via their import and export, and more importantly, via their synthesis and degradation (Figure 1).

Cellular import of polyamines is mediated via the action of multiple transporters [20]. Polyamine transporters are not definitively characterized in mammals. However, several studies indicate that polyamines are imported via the organic cation transporters (OCT) 1–3 (SLC22A1-3) and SLC12A8A, as well as through endocytosis, while the export of acetylated polyamines occurs through SLC3A2 [20,21,22,23,24]. Polyamine biosynthesis is dependent on the conversion of arginine to ornithine. Ornithine decarboxylase (ODC)-mediated decarboxylation of ornithine leads to the generation of putrescine. Sequential enzymatic addition of aminopropyl groups to putrescine and spermidine by spermidine synthase and spermine synthase, respectively, leads to the formation of spermidine and spermine. Polyamines are degraded through their back-conversion via the spermidine/spermine N^1^-acetyltransferase/N^1^-acetylpolyamine oxidase (SAT1/PAOX) cascade and the direct oxidation of spermine by SMOX. Oxidation of acetyl-spermine and acetyl-spermidine by PAOX and spermine by SMOX generates toxic molecules such as H_2_O_2_ and reactive aldehydes [25]. Recent studies indicate that mutations that alter the activity of enzymes involved in polyamine metabolism (e.g., mutations in ODC or spermine synthase) severely affect the growth and development. [26,27,28].

## 3. Expression and Activity of Polyamine Catabolic Enzymes, SAT1, and SMOX Increases in Injured Tissues

Polyamine catabolism is enhanced in the kidney, brain, and liver in response to I/R, toxic, septic, and traumatic insults, and in stomach and colon in response to bacterial infections [4,10,29,30,31,32]. In addition, studies have shown that the expression of polyamine catabolic enzymes increases in remote organs (such as the liver) following an initial acute injury to the kidney [33]. Enhanced catabolism of polyamines, as evidenced by increases in SAT1 and SMOX expression and their oxidation, have been demonstrated in the brain after I/R, traumatic, and excitotoxic injuries [10,34,35,36,37].

In addition, studies by Tomitori et. al. demonstrated the presence of increased levels of PAOX, SMOX, and acreolin in the plasma of stroke patients, suggesting the potential role of these molecules as biochemical markers of this injury [38]. Enhanced polyamine oxidation was also observed in excitotoxic and ischemic retinopathies [39,40]. The expression of both SAT1 and SMOX is also elevated in the myocardium, as seen in a model of cardiac ischemic reperfusion (I/R) injury (Figure 2).

Enhanced polyamine catabolism and derangements in polyamine levels have been documented in the livers of animals subjected to hepatic I/R and CCl_4_-induced hepatotoxic injuries [6,9]. This was illustrated by increases in SAT1 and SMOX expression and activity, heightened putrescine, and reductions in both spermidine and spermine levels [9]. Using global *Sat1*-KO and hepatocyte specific *Sat1*-KO mice, it was further demonstrated that inactivation of the *Sat1* gene significantly reduces the severity of liver damage in both I/R and CCl_4_-induced hepatic injuries [9,32]. The mechanism by which enhanced polyamine catabolism and derangements in polyamine levels lead to hepatic injury are not completely understood; however, studies by Hyvonen et al. showing that α-methylspermidine treatment reduces the severity of hepatic and pancreatic injuries provides a possible clue [41,42]. In the pancreas, this protection may be due to a combination of stabilization and prevention of degranulation of zymogen granules by polyamine mimetics and cathepsin B release in response to oxidative stress. In addition, α-methylspermidine can both reduce the severity of tissue injury via reducing the inflammatory response, as well as the induction of polyamine catabolic enzymes [41,42]. Spermine has also been shown to reduce the severity of I/R-mediated cardiomyocyte injury via the enhancement of autophagy [43].

The ablation of the *Sat1* gene in mice leads to altered fat metabolism and increased white fat and reduced brown fat content, as well as the dysregulation of beige adipocyte biogenesis [44,45]. Recent studies by Jain et al. also demonstrate that co-treatment of animals with inhibitors of SMOX (e.g., MDL72527) and SAT1 (e.g., diminazene aceturate) leads to bronchial epithelial cell injury [46]. These studies suggest that polyamine back conversion (SAT1-mediated polyamine degradation) and oxidation are important in normal physiological conditions.

## 4. Expression and Activity of Polyamine Catabolic Enzymes in AKI

Expression of SAT1 and SMOX increases in kidneys of animals subjected to AKI caused by I/R, endotoxin administration, or cisplatin treatment [4,7,8,32,47]. Our studies have shown a similar increase in the expression of these polyamine catabolic enzymes in kidneys of patients with a delayed graft function, a model of acute tubular necrosis post kidney transplant. Studies by Pirnes-Karhu et al. showed that in LPS (lipopolysaccharides)-induced injury, *SAT1* over-expressing transgenic mice had a more profound anti-inflammatory response even though their mortality was similar to wild type (Wt) animals [48]. Although these results suggest that enhanced polyamine catabolism may be protective, the more likely explanation is that the less severe inflammatory response may be due to pre-conditioning resulting from the existing oxidative stress in response to the elevated basal expression and activity of SAT1 [5,49,50,51,52,53]. The protective role of pre-conditioning via the induction of mild oxidative stress is well established in acute liver and kidney injuries [54,55]. In mice, the ablation of the *Sat1* and *Smox* genes reduces the extent of tubular injury and preserves kidney function after I/R, endotoxic, and nephrotoxic injures [7,8,32,47]. In I/R injury, the ablation of *Sat1* in proximal tubule epithelial cells, the targets of I/R injury, also imparted protection against tubular damage and preserved kidney function [47]. The maladaptive role of SMOX in I/R-, endotoxin-, and cisplatin-induced-AKI was also confirmed using either spermine oxidase knockout (*Smox*-KO) mice or through the inhibition of polyamine oxidases by MDL72527 [7].

The exact molecular basis of the enhanced expression of polyamine catabolic enzymes, SAT1, and SMOX in injured tissues remains unknown. In vitro studies indicate that the expression of *SAT1* and *SMOX* transcripts increase in response to oxidative stress induced by H_2_O_2_ or as a result of treatment with pro-inflammatory cytokines, such as tumor necrosis factor-α (TNF-α) and interleukin-6 (IL-6) [56,57,58,59]. The aforementioned molecules are also important mediators of tissue damage in AKI [60,61,62,63]. The induction of SAT1 gene expression in response to H_2_O_2_ inducers and TNF-α is in part mediated by the activation of nuclear factor κB (NFκB) [57,59], a key transcription factor involved in the mediation of maladaptive immune/inflammatory changes in AKI [64,65]. There also appears to be a regulatory cross-talk between *SAT1* and *SMOX* genes, where increased expression of enzymatically active SAT1, but not its inactive counterpart, leads to the induction of *SMOX* expression in HEK293 cells that are stably transfected with Tetracycline-inducible Wt, stable or inactive SAT1 expression constructs (Figure 3). The responsiveness of both *SAT1* and *SMOX* genes to IL-6 and TNF-α, which are important mediators of changes that contribute to tubular damage in AKI, suggests that their enhanced expression in the AKI of various etiologies may in part be due to the elevated levels of these cytokine. At the cellular level, the role of tumor protein p53 (p53), a molecule that plays a maladaptive role in the tubular epithelial cell damage in both I/R and cisplatin injuries [66,67,68], in the induction of both SAT1, and the activation of ferroptosis, has been documented [69]. This upregulation of SAT1 may also contribute to the induction of SMOX. These findings suggest the presence of a multi-layered mechanism whereby an initial hypoxic or toxic injury leads to the activation of P53, which may then enhance the expression of polyamine catabolic enzymes. In more advanced stages of AKI, the increased expression of TNF-α, a known activator of NFκB and p53, may lead to the persistence of the polyamine catabolic activity via the NFκB and p 53 mediated expression of SAT1 and further enhancement of polyamine catabolism.

## 5. In Vivo and In Vitro Effects of Enhanced Polyamine Catabolism

The adverse effects of increased polyamine catabolism, specifically increased SAT1 expression, have been confirmed in both transgenic animals and cultured cells [5,70,71]. Transgenic expression of SAT1 leads to loss of hair, skin lesions, and female infertility [70]. Additionally, treatment of SAT1-transgenic mice with N^1^,N^11^-diethylnorspemine, an inducer of polyamine catabolism, led to a significant elevation in polyamine back-conversion and is accompanied by a mortality rate of nearly 50% [72]. Transgenic over-expression of SMOX had no effect on the expression and activity of other polyamine pathway enzymes and only led to increased tissue spermidine levels without affecting the tissue content of other polyamines [73]. These mice did not exhibit any overt phenotypic changes, partly as a result of an elevated antioxidant response that offset the production of cytotoxic molecules as a result of the enhanced SMOX activity [73]. However, specific overexpression of SMOX in the cerebral cortex led to increased susceptibility to excitotoxic injury [34].

The over-expression of SAT1 in HEK293 cells led to reductions in spermine and spermidine levels, disruption of cell cytoskeleton and adhesion, and reduced cell proliferation [5,12]. Several derangements, such as the induction of DNA damage and DNA repair response, onset of endoplasmic reticulum stress, and mitochondrial dysfunction were observed in SAT1-overexpressing cells [5,7,47,71]. These changes recapitulate many of those that occur in the renal tubular epithelial cells during AKI [5,7,47]. Mandal et al., using an adenovirus expression system, demonstrated that enhanced expression of SAT1 leads to the depletion of spermine and spermidine [12]. In these studies, the increased polyamine catabolism and alterations in polyamine levels led to the disruption of protein synthesis and eventual growth arrest via the inhibition of initiation and progression of protein synthesis, which is characterized by a loss of polysomes and accumulation of monosomes [12,74]. The latter changes in polysome to monosome ratios are similar to that described by Landau et al. [75], who showed that the depletion of polyamines and reduction in hypusinated eIF5A lead to the accumulation of polysomes and a similar reduction in protein synthesis. The decline in cellular levels of hypusinated eIF5A was also demonstrated in SAT1-expressing HEK293 cells undergoing endoplasmic reticulum stress and an unfolded protein response [7].

## 6. Biochemical Basis of Renal Injury Caused by Enhanced Polyamine Catabolism

Polyamines are known scavengers of hydroxyl radicals, and modulate the generation of reactive oxygen intermediates [76,77,78]. Specifically, spermine has been shown to scavenge free radicals, to chelate transition metals, such as iron, and to modulate Fenton reactions [79]. Studies by Kim demonstrated that the provision of spermidine in the drinking water reduced the severity of I/R-induced kidney injury in mice through the inhibition of DNA nitration and modulation of poly (ADP-ribose) polymerase 1 (PARP1) activity [80]. The above studies suggest that reductions in cellular polyamine levels and diminished anti-oxidant capacity as a result of their enhanced degradation may play a role in the maladaptive effects of enhanced polyamine catabolism in acute injuries. Polyamines are also present in significant intracellular concentrations (millimolar range); and therefore, substantial concentrations of H_2_O_2_ and reactive aldehydes (e.g., 3-aminopropanal, 3-acetoaminopropanal, and acrolein) can be produced via their catabolism [81]. While H_2_O_2_ through the generation of hydroxyl radicals causes DNA lesions [82,83], aminoaldehydes can disrupt the integrity of lysosomal and mitochondrial membranes, causing cellular injury through activation of necrotic and apoptotic pathways [84,85,86,87]. Therefore, while the reductions in polyamine levels compromise the ability of cells to fight oxidative stress-induced damage, the production of toxic metabolites through enhanced polyamine degradation can also lead to cell injury via the induction of oxidative stress and organelle damage.

Enhanced polyamine catabolism in AKI can lead to tubular epithelium damage via derangements in polyamine levels, generation of toxic products of polyamine catabolism, or both. Comparison of renal polyamine levels in I/R, sepsis and cisplatin-mediated AKI revealed there is a significant increase in renal putrescine levels, while only in cisplatin-mediated AKI, is there a significant decrease in renal spermine levels [4,7,8]. Although these findings do not completely rule out the potential role of polyamine depletion as a contributing factor in I/R and endotoxin-induced AKI, they emphasize the importance of derangements in putrescine and spermine levels in the mediation of cisplatin-induced AKI. An important note regarding the role of alterations in polyamine levels in the mediation of AKI is that in all such studies total polyamine levels were compared in the injured versus uninjured kidneys; however, the heterogeneity of renal epithelial cell populations and their differential susceptibility to I/R and toxic insults are not accounted for in such measurements. Future studies may benefit from comparing polyamine levels in injured tubular epithelia versus unaffected tubular epithelia of the same region of injured and uninjured kidneys in order to yield more precise and pertinent results regarding the role of derangements in cellular polyamine levels in AKI.

Due to the equivocal results of the effect of AKI induced alterations in renal polyamine levels, the role of H_2_O_2_ and reactive aldehydes (e.g., 3-acetoamidopropanal, 3-aminopropanal and acrolein) generated as a result of enhanced polyamine catabolism in the mediation of cell injury becomes more pertinent, especially because of high cellular polyamine levels. The role of toxic products of polyamine degradation in the mediation of AKI were established in in vitro studies that showed that the neutralization of H_2_O_2_ in cultured cells expressing high levels of SAT1 resulted in the reduction of the severity of cell injury [5]. These observations were further confirmed in vivo by demonstrating that the neutralization of H_2_O_2_ or reactive aldehydes diminished the extent of renal tubular damage and the severity of kidney dysfunction in cisplatin-mediated AKI [7].

## 7. Mechanistic Basis of Cell Injury and Tissue Damage Caused by the Upregulation of Polyamine Catabolism

Increased polyamine catabolism in cultured cells leads to reduced cell proliferation [5,71,74] and the onset of apoptosis [47]. In vitro, over-expression of SAT1 can induce both single-stranded or double-stranded DNA breaks and activate the DNA damage/repair response proteins, such poly(ADP-ribose) polymerase, ataxia telangiectasia mutated, and ataxia telangiectasia and Rad3-related protein [47,71]. This cascade of cellular events in turn activates the cell cycle check point protein kinases, CHK 1 and 2, and through modulation of their downstream targets, regulates the progression of the cell cycle leading to a G2 to M transition arrest [71]. Increased expression and activity of SAT1 in cultured cells also leads to the loss of mitochondrial integrity and activates the endoplasmic reticulum stress/unfolded protein response (ERS/UPR) [7,47]. The severe and ongoing cellular stress brought about by the above changes also leads to the induction of apoptosis [7,47].

Multiple pathways and responses contribute to the pathogenesis of AKI. The loss of mitochondrial integrity and activation of endoplasmic reticulum stress/unfolded protein response (ERS/UPR) as a result of oxidative stress are changes whose roles in the induction of cell and tubular damage in AKI are well documented [88,89,90]. Reactive oxygen species and reactive aldehydes, which are molecular mediators of tissue damage generated as a result of polyamine degradation, are important inducers of DNA lesions, mitochondrial damage, and ERS/UPR, all of which are known triggers of apoptosis in tubular epithelial cells and tubular damage in AKI [90,91,92]. The activation of the inflammatory/innate immune response has also been identified as an important factor in the induction of AKI [93,94]. Our studies indicate that the ablation or inhibition of SMOX and SAT1, as well as neutralization of the products of their enzymatic activity (e.g., H_2_O_2_ and reactive aldehydes), reduce the severity of endotoxin-, I/R-, and cisplatin-induced AKI by dampening the severity of oxidative stress, the inflammatory/innate immune response, and the ERS/UPR response [7,47].

## 8. Conclusions

Studies have shown that polyamine catabolism, via alteration of polyamine levels and the production of toxic metabolites, contributes to the pathogenesis of a variety of injuries, including those of AKI of different etiologies. The enhanced catabolism of polyamines in cells can result in DNA damage, mitochondrial dysfunction, and ERS/UPR, all of which can lead to the onset of tubular epithelial cell damage and death. Through defining the chemical and mechanistic basis of polyamine derangements, as well as identifying the causes for increased polyamine catabolism, several novel therapeutic options have emerged that can be targeted for the treatment of acute injuries of the kidney, as well as other organs. Such treatments can directly target the activity of SMOX (e.g., MDL72527) or SAT1 (e.g., diminazine aceturate), neutralize their cytotoxic products (e.g., *N*-2-mercaptopropionyl glycine), or use stable mimetics of polyamines (e.g., α-methylspermidine) in order to reduce the inflammatory response and to prevent oxidative injury. It also should be noted that long-term treatments with the inhibitors of polyamine catabolic enzymes or ablation of the SAT1 gene can lead to pulmonary injury and altered fat metabolism [44,45,46]; hence, co-treatment with SAT1 and SMOX inhibitors or long-term inhibition of SAT1 may have unexpected adverse effects.

Evidence thus far indicates that while deactivation or inhibition of polyamine catabolic enzymes and neutralization of the toxic products of their degradation reduce the severity of AKI in experimental animal models, they do not impart complete protection. Further studies are needed to determine the therapeutic efficacy of the above in combination with other maneuvers in the treatment of AKI in animal models. These studies should lead to the development of treatment protocols that can be tested in AKI patients and ultimately lead to the development of novel and more effective therapeutic interventions.

## Figures and Tables

**Figure 1 ijms-20-04790-f001:**
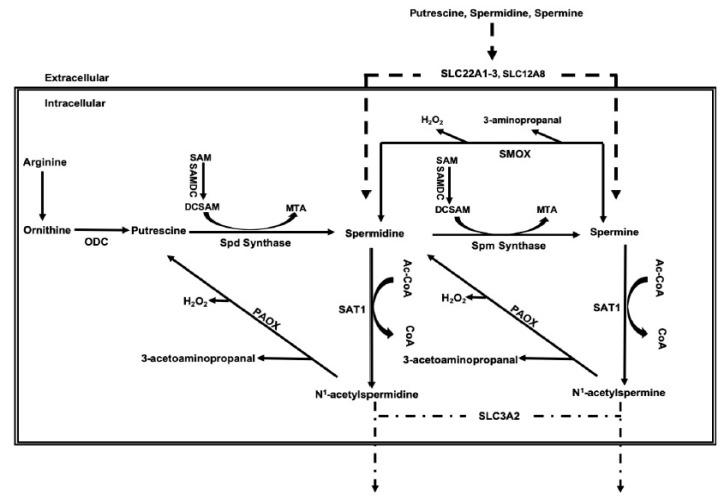
Schematic presentation of the metabolic pathway and transporters that regulate the cellular polyamine levels.

**Figure 2 ijms-20-04790-f002:**
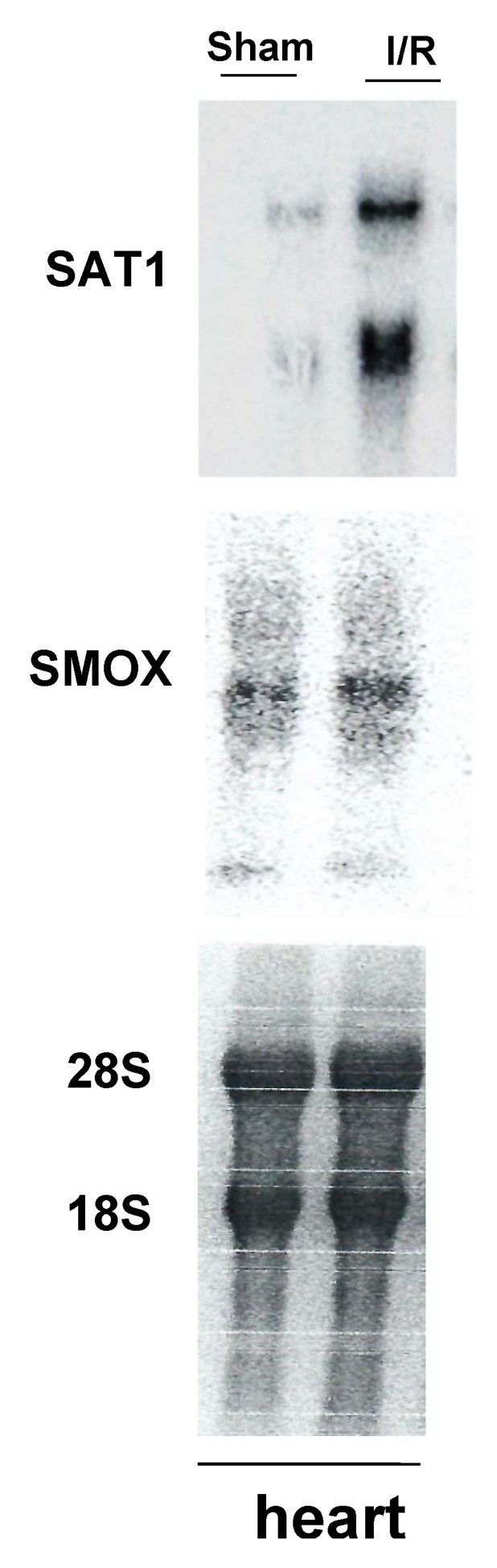
Effect of cardiac ischemia/reperfusion (I/R) injury on polyamine catabolism. Cardiac ischemia was induced in mice via the transient (30 min) occlusion of the left anterior descending artery. Cardiac tissue was harvested at 24 h post-surgery, processed for RNA extraction, and subjected to northern blot analysis (30 μg of RNA/lane). The results indicate that the expression of transcripts for spermine/spermidine N^1^-acetyltransferase (*Sat1*) and spermine oxidase (*Smox*) genes increases following cardiac I/R injury. The northern blot shown here is a representative of four independent sample pairs. The 28S and 18S rRNA (bottom) show comparable RNA loading in sham and I/R samples.

**Figure 3 ijms-20-04790-f003:**
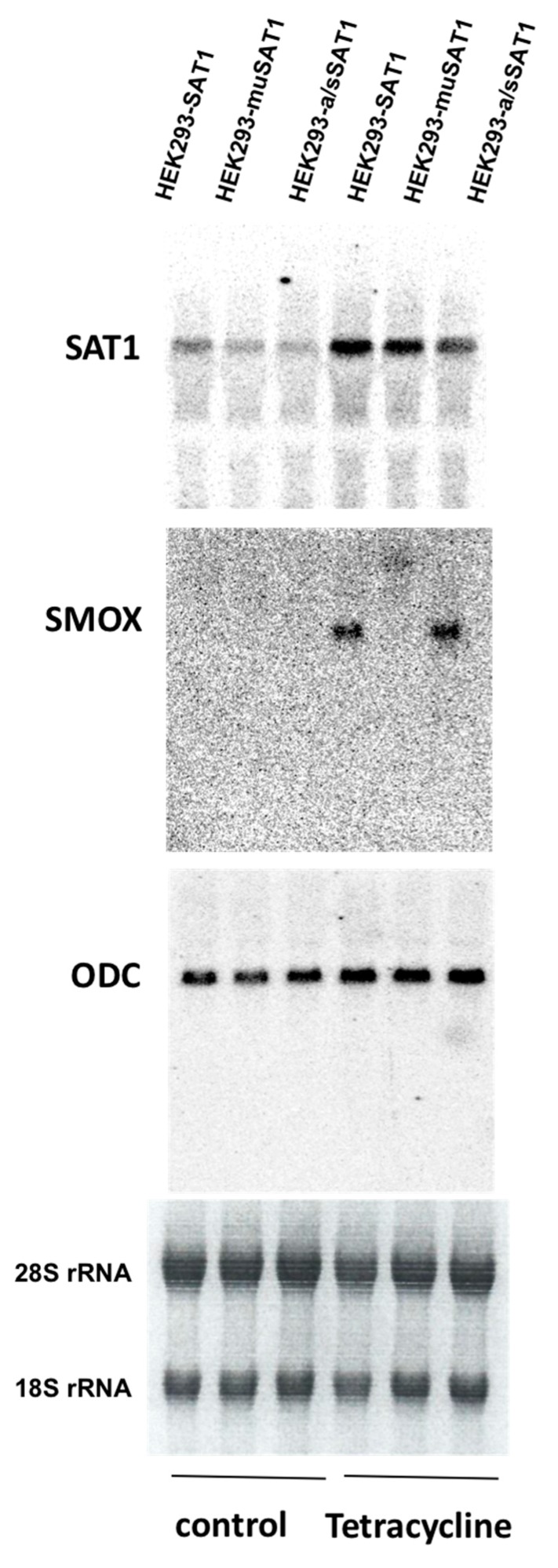
Transcriptional cross-talk in the polyamine metabolic pathway. The expression levels of spermidine/spermine N^1^-acetyltransferase (SAT1), spermine oxidase (SMOX), and ornithine decarboxylase (ODC) were compared in HEK293 cells that expressed either wild type (Wt) (HEK293-SAT1), inactive (HEK293-muSAT1), or an isoform of active/stable SAT1 with a greater enzymatic activity (HEK293-a/sSAT1) under the control of tetracycline-inducible promoter. Total RNA was extracted from tetracycline-induced (Tetracycline) and un-induced (control) cells at 24 h and subjected to northern blot analysis. Examination of SAT1, SMOX, and ODC mRNA levels indicated that the induction of enzymatically active SAT1 by tetracycline (lanes 1 and 3 from right) significantly enhances the expression of SMOX without impacting the expression of ODC. The expression of SMOX remained undetected in cells not treated with tetracycline (lanes 1 to 3 from left; also designated as control). The 28S and 18S rRNA (bottom) show comparable RNA loadings in the control and tetracycline-treated samples.

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
