# Peer review of "Polyamine Catabolism in Acute Kidney Injury"

_ijms, 2019, doi:10.3390/ijms20194790_

Round 1

Reviewer 1 Report

In Figure 3, renal biopsy samples were used. It is not clear to me if this review article required any IRB, but that should be mentioned in the article.

In Figure 3, the scale or magnification should be described.

Character corruptions were found throughout the manuscript (Lines 125, 164, 168, 173, 182, and 311), so the authors should corrected them.

In Figure 5, the authors should describe the precise name of cell line and conditions of the experiment in the figure legend and manuscript (Lines 273-275).

As for multiple pathways and responses contribute to the pathogenesis of AKI, ROS plays an important role. The authors should add a reference (Int J Mol Sci. 2016 Nov 1;17(11). pii: E1826) in the description (Lines 291-296).

Author Response

We thank the reviewer for their constructive and astute comments and have amended the manuscript according to the reviewers' suggestions.

The samples used in Fig. 3 were obtained from a library of educational specimens and as such did not require an IRB. As per reviewers' suggestion, we have included the magnification of images in the figure legend. Corrupted characters were corrected through out the text. As per reviewers suggestions we have added the name of each cell line and additional information about the experimental conditions to the figure legend. The recommended IJMS article was referenced and cited.     

Reviewer 2 Report

The manuscript is an excellent/comprehensive review that describes the role of polyamine catabolism in various acute tissue injuries with a specific focus on AKI. It discusses the biochemical and mechanistic basis of tissue damage by hydrogen peroxide, amino-propionaldehyde and acrolein generated by activation of polyamine catabolic enzymes, N1-spermidine/spermine acetyltransferase (SAT1), polyamine oxidase (PAOX) and spermine oxidase (SMOx) in the injured tissues. The potential molecular pathway leading to transcriptional activation of expression of SAT1 and SMOX may involve NFkB, TNFa and P53. The tissue damages by reactive oxygen species generated by polyamine catabolic enzymes may result from DNA lesions, mitochondrial damage and ERS/UPR. The possibility of inhibition of SAT1 and SMOX in potential therapy of AKI is discussed. The manuscript will be strengthened by addressing the following points.

The data shown in Figs 2 -5 need more specific descriptions in the legends. If the original data was presented previously, cite the original paper by adding (modified from ref X) in the legends. For example, Figs. 4 and 5 should indicate that SAT1 was expressed under the tet-inducible promoter and it would be better to include SAT1 Northern as a top panel. P2, line 66-70, regarding to a general description of polyamine function in cells, the following points should be included. A majority of cellular polyamines is bound to RNA and their most important cellular function is the regulation of translation (Igarashi, K., and Kashiwagi, K. (2015) Modulation of protein synthesis by polyamines. IUBMB Life 67, 160-169, and ref 50, Mandal et al). There is not enough mention of the role of PAOX in the generation of reactive oxygen species and tissue damage. Although PAOX is considered to be constitutively expressed, the induction of PAOX was shown to be increased in IRI kidney in ref 86 (Zahedi etal 2003). A high accumulation of putrescine and N1-Ac-Spd in 293T cells overexpressing SAT1 (ref 50) suggests that N1-Ac-Spd is not rapidly excreted but is converted to putrescine by APAO (this extent may vary with cells and tissues). Furthermore, polyamine oxidase and acrolein were reported as biomarkers of stroke. (Hideyuki T. et al, 2005, Stroke, 36: 2609-2613). Line 75, polyamine transporters are not well/definitely characterized in mammalian cells. Line 142, a comma should be inserted “-induced injury, SAT1 over-….” P5, correct the names with the correct symbols; TNF-a, NFkb, etc. Line 168, “are” should be “is” Line 172, “active but not inactive SAT1” is confusing. “active SAT1 (but not inactive SAT1)” may be better. Line 194, two references (Mandal et al 2013, Mandal et al 2015, …) should be deleted. Line 206, it was not “Inducible” expression in (Mandal et al 2013 ), but overexpression by AdSAT1 transduction. Delete “putrescine” from “ reductions in putrescine…” as SAT1 overexpression depleted Spd and Spm but enhanced Ptc. Line 218, the correct reference is (Mandal et al, 2013), not (Mandal et al, 2015). Line 220, Correct to “eIF5A” here and other places in the manuscript. Line 245, change “polyamine synthesis” to polyamine catabolism” Line 311, correct to “a-methylspermidine” here and other places.

Author Response

We appreciate the referee's astute observations and comments. The following is an outline of changes that were made to the manuscript based on the referee's recommendation.

All data shown in Figures 2-5 are original and unpublished. As per reviewers' suggestions we have expanded the figure legends to include the tet-inducible nature of the construct and have included the northern blot image of SAT1 expression. (figure legends, Figs. 4 and 5) A statement concerning the interaction of polyamines with RNA and their role in the modulation of protein translation has been added to the manuscript and the appropriate articles have been referred to and cited. (Lines 69-70) The polyamine oxidase referred to in Zahedi et.al., 2003 was the induciblePAO-1 which was cloned and characterized by Wang et.al., in their 2001 Cancer Research manuscript and not acetylpolyamine oxidase (PAOX). The name of this inducible polyamine oxidase has since been changed from PAO-1 to PAOh1/SMO to SMO and finally to its current designation SMOX (spermine oxidase). As the reviewer correctly indicated PAOX is a constitutively expressed peroxisomal enzyme. The "polyamine oxidase" designation in our 2003 manuscript was based on the nomenclature at the time of publication. We agree with the reviewer that the increase in PUT in our tet-inducible system may in part be due to the activity of PAOX that can be released as a result of oxidative injury-induced peroxisomal disruption, but neither our nor other studies have examined the effect of increased polyamine catabolism on the integrity of peroxisomes and the release of PAOX from these organelles. Based on the article by Hideyuki et. al., 2005 we have added additional discussion regarding the potential role of serum SMOX, PAOX and acrolein as markers of stroke. (Lines 107-109) A statement regarding the definite characterization of polyamine transporters has been added to the manuscript. (Lines 77-78) All corrupted symbols have been corrected. The word "are" has been changed to "is."  (Line 177) The statement "active but not inactive SAT1..." has been amended. (Lines 181-183) References were deleted as per reviewers' suggestion. (Lines 209-210) "Inducible expression" has been changed to "Over-expression." (Line 220) "Putrescine" has been deleted. (Line 220) Mandel reference has been corrected. (Line 222) "EIF5alpha" has been replaced with eIF5A. "Polyamine synthesis" has been changed to "polyamine catabolism." All corrupted symbols have been corrected.   

Round 2

Reviewer 1 Report

In figure 3, the author used "renal biopsy," and which is "using educational samples provided by the Department of Pathology and Laboratory Medicine, university of Cincinnati Medical School. However, even if this biopsy sample was permitted by education, it is necessary to be permitted by ethical committee in order to use it for publication. If the authors can't, Figure 3 should be deleted. 

Say again, ethical committee is needed in research use of biopsy sample.  

Reviewer 2 Report

The authors have revised the manuscript by deletion of two old figures (Figs. 3 & 5) and by modification of the two Figures (new Figs. 2 & 3) and satisfactorily addressed the points raised.

It is not critical, but the authors may consider arranging the panels of the new Figs. 2 & 3, horizontally (instead of vertically) to save space and/or mark the RNA sizes next to the bands of interest or include them in the legend. 

Round 3

Reviewer 1 Report

The authors responded to the reviewers' comments carefully.